# Is absorptive capacity the "panacea" for organizational development? A META analysis of absorptive capacity and firm performance from the perspective of constructivism

Kun Pu[ID]*◷, Wei Liu◷

School of Economics and Business Administration, Chongqing University, Chongqing, PR China

◷ These authors contributed equally to this work.
* 20080329@cqu.edu.cn

**Data Availability Statement:** All relevant data are within the manuscript and its Supporting Information files.

## Abstract

There is a long-standing academic consensus that the higher the absorptive capacity of an organization is, the better its performance. Recently, however, the assumption that absorptive capacity can unconditionally contribute to firm performance has begun to be challenged, and empirical results differentiating absorptive capacity and firm performance have also begun to emerge. Therefore, to effectively integrate the variability of different empirical results and reveal the mechanism by which absorptive capacity acts on firm performance, this paper explores the relationship between absorptive capacity and firm performance from the perspectives of both theoretical exploration and META analysis. Through the process of theoretical combing, this paper finds that the existing core concept of absorptive capacity is based on cognitivism, and the existing process behind absorptive capacity is based on a linear cognitive information processing process that focuses on the internal cognitive structure and process of the subject. However, due to the dynamic and complex nature of social phenomena, the process model cannot effectively reflect the influence of contextual factors on their relationships. Next, based on the results of theoretical sorting, the results of existing empirical studies are synthesized by means of META analysis and different contexts are examined, finding that the role of absorptive capacity on firm performance has significant contextual characteristics, among which the research context, economic context and sample context all have significant but distinct moderating effects on absorptive capacity and firm performance. Overall, by including contextual factors, this paper further deepens the understanding of the relationship between absorptive capacity and firm performance. It also provides a preliminary basis for the role of contextual factors in absorptive capacity.

## Introduction

Currently, knowledge has become the core competitive resource of companies [1,2]. The closed knowledge development system characterized by the R&D activities carried out within an organization is gradually changing to an open knowledge development and innovation

**Funding:** The author(s) received no specific funding for this work.

**Competing interests:** The authors have declared that no competing interests exist.

system characterized by the management of knowledge flow and search for knowledge sources outside the organizational boundary [3–5]. The rapid changes in technology and the diversity of market demands have also increased the levels of external knowledge resources that are required by an organization [6,7]. These trends implicitly require firms to increase the effectiveness of their external knowledge acquisition, and the ability of an organization to acquire, transform, and commercialize external knowledge is exactly what is revealed by absorptive capacity [8]. Therefore, the general consensus among existing research is that when organizations have better absorptive capacity, organizations can better identify external knowledge, transform and utilize that knowledge faster, and thus take steps toward better performance [9–11]. However, can companies truly improve corporate performance by continuously improving their absorptive capacity? Thus far, the empirical research results on the relationship between absorptive capacity and firm performance do not explain this issue very well. For example, a large number of mainstream empirical studies have verified a positive linear relationship between absorptive capacity and firm performance [12–14]. In contrast, Wales, Parida [15] found an inverted U-shaped relationship between absorptive capacity and firm performance through empirical testing, i.e., there are some samples that are negatively correlated with corporate performance. Wales, Parida [15] believe that this is because the cost of knowledge acquisition is greater than the benefits it can bring. Ben-Oz and Greve [16], Larrañeta, Galán [17] directly conclude that absorptive capacity is negatively correlated with firm performance under certain circumstances. Although this difference in empirical results will lead to the fragmentation and superficialization of the theory and hinder the understanding of absorptive capacity theory [18], the differentiation also provides the conditions for a more in-depth study of the characteristics of absorptive capacity and the discovery of the mechanism through which absorptive capacity impacts firm performance.

Context is extremely important in management research because companies make decisions and take actions in specific contexts [19]. Especially in the knowledge acquisition activities of an organization, context is a key element in understanding knowledge, and, at the same time, context also affects the characteristics and results of a company's knowledge activities [20–22]. Nevertheless, the impact of contextual factors on absorptive capacity is rarely mentioned in the literature. By reviewing and summarizing the literature on absorptive capacity, this paper finds that the core connotation of absorptive capacity is based on the cognitivist learning view, which holds that in the process of cognition, the environment only provides stimulation for cognition, and then these stimuli and the subject's existing cognitive structure work together to determine the subject's behavior. The subject's cognitive structure and cognitive process are at the core of learning. When Cohen and Levinthal [8] discuss the concept and connotation of absorptive capacity, they make inferences based on individual cognition, which involves a large number of cognitivist viewpoints and theories [23]. The process model of absorptive capacity [24,25] also follows the information processing theory of cognitivism, which posits that learning is a procedure of processing, storing and extracting information, and the information processing model for computers is used to infer psychological processes. However, social phenomena are complex and dynamic, and the concept of absorptive capacity that emphasizes natural attributes ignores the differences engendered by changes in organizational context in explaining the relationship between absorptive capacity and firm performance.

Thus, to better answer the questions that arise from the differences between the empirical results on absorptive capacity and those on firm performance, this paper introduces the concepts of knowledge and learning to constructivism and discusses the relationship between context and knowledge and that between context and learning. Subsequently, the paper further discusses the theoretical connection between absorptive capacity and organizational context.

In particular, the traditional knowledge-based view holds that knowledge is a resource, so knowledge is completely objective [26]. However, from the perspective of constructivism, knowledge is seen as a social construction in work practice. In this view, knowledge exists on the basis of specific meanings and contexts. Therefore, it has obvious social attributes. Based on this view, meaningful knowledge is deeply embedded in daily work practice and circumstances. The acquisition of new knowledge comes from the social construction of work practice [27]. Furthermore, constructivism emphasizes that human reason is always embedded in concrete situations because thinking relies on concrete, connected contextual relationships to provide meaning rather than relying on abstract processes that are isolated and disconnected from concrete content [28]. Therefore, from a constructivist perspective, absorptive capacity is not only the mechanical processing and acquisition of information but also reflects the way in which practices and contexts shape organizational knowledge. In summary, this paper theoretically clarifies the context-dependent characteristics of absorptive capacity. In addition, to quantitatively analyze the source of the difference between absorptive capacity and firm performance, this paper uses META analysis to code and test the results of existing empirical literature to determine how different types of contexts moderate the results of absorptive capacity and firm performance.

## Literature review

### Absorptive capacity process model

Cohen and Levinthal [8] define absorptive capacity as the ability of an organization to identify the value of external information, transform it, and apply it for commercial ends. Through the induction and inference of the cognitive model, the author hypothesizes that the core factor affecting an organization's absorptive capacity is their existing knowledge base. When an organization has a rich knowledge base, it can more efficiently acquire new external knowledge and effectively apply it. Subsequently, a large number of scholars have begun to refine the concept of absorptive capacity, and the theory of absorptive capacity is becoming increasingly prevalent. [29–31]. One of the most influential refinements is the absorptive capacity process model proposed by Zahra and George [24], in which the authors divide absorptive capacity into four dimensions: acquisition, assimilation, transformation, and exploitation. At the same time, acquisition and assimilation are separately grouped into potential absorptive capacity, and transformation and exploitation are grouped into realized absorptive capacity. Zahra & George propose that potential absorptive capacity tends to affect a company's strategic flexibility, which in turn affects their long-term performance, while realized absorptive capacity can more directly improve short-term performance, but it easily falls into the "capacity trap." In addition, some scholars divide absorptive capacity into a process model containing three dimensions. For example, Lane, Koka [25] linked the dimension of absorptive capacity with learning and proposed a process model of three dimensions: exploration learning, transformative learning and exploitation learning. Subsequent theoretical and empirical research has been developed almost entirely by using absorptive capacity as a process model of knowledge acquisition [32–34]. Overall, the process-based absorptive capacity model effectively explains the internal mechanism and related important factors of an organization's acquisition of external knowledge.

However, after reviewing the empirical literature on the relationship between absorptive capacity and firm performance, inconsistent empirical results are found. For example, a large number of empirical studies support the conclusion that improving firm absorptive capacity directly improves firm performance [14,31,35]. From the perspective of organizational learning, absorptive capacity can indeed enhance the transfer of knowledge, theoretically promote

innovation, and thus create a sustainable competitive advantage. However, when considering that an improvement of absorptive capacity may lead to an increase in an organization's costs, an inverted U-shaped relationship between absorptive capacity and firm performance appears. Wales, Parida [15] used a sample comprising small-scale technology companies to verify the above relationship. Moreover, in certain cases, such as that of technology start-ups, there is even a decrease in firm performance detected as absorptive capacity increases [16,17].

In summarizing the extant literature, this paper finds that the empirical research on absorptive capacity and firm performance has obvious context-dependent characteristics, i.e., the results of absorptive capacity and firm performance under different contextual conditions are quite different. However, the concept of absorptive capacity was based on individual cognitive psychology when it was first proposed [8], and the subsequent process model of absorptive capacity is based on information processing learning theory. The focus of previous studies has been on the influence of the subjects' cognitive structure on the learning process. Many scholars have criticized that view, however, such as Bruner, J. [36], who claimed that information processing theory focuses on information processing and cognitive structure while ignoring the meaning of informational content to the subject and the key role of context in meaning construction. Therefore, if an empirical test of absorptive capacity and firm performance is conducted without accounting for contextual factors, then the results will contain discrepancies. In addition to the specific situation faced by an organization, the researcher's setting for assessing absorptive capacity and firm performance, that is, the research context, also affects the relationship between absorptive capacity and firm performance. For instance, Cohen and Levinthal [8] proposed that absorptive capacity is reflected in the existing knowledge base of an enterprise, and the R&D expenditure of the enterprise is then used as a method to measure absorptive capacity. Since Jansen, Bosch [32] believe that absorptive capacity is reflected as dynamic capacity, it is necessary to measure absorptive capacity by means of scales. This measurement difference can make a large difference in empirical results. Likewise, different contexts can produce different feedback in regard to different firm performance measures.

On this basis, two questions are proposed to be behind the difference between absorptive capacity and firm performance: (1)How does the organization's context moderate the relationship between absorptive capacity and firm performance, and (2) how does the research context moderate the relationship between absorptive capacity and firm performance? To answer the above questions, this paper introduces the concept of knowledge and learning in constructivism and discusses the mechanism of interaction among knowledge, learning and the environment in constructivism. At the same time, the connection between organizational absorptive capacity and context is established.

## Constructivism and context dependence

Constructivism is one of the most influential theories and trends of thought to emerge since the 1960s, and it provides an important theoretical basis for today's social and educational reforms. Similarly, the ideas within constructivism have been increasingly accepted by scholars of strategic management and organization theory in recent years [37,38]. Contrary to realism, which holds that theory exists independently of people, constructivism holds that theory is an active construction of people within a specific context. As Glaserfield [39] has said, "to the constructivist, concepts, models, theories and so on are viable if they prove adequate within the context they were created." Realism posits that the research process is similar to the excavation process and that meaningful ideas exist in the soil of the phenomenon. Constructivism claims that the research process is similar to sculpture, and the theoretical basis of human beings creates reality, i.e., knowledge, in the process of sculpting. From this perspective, knowledge

combines the subjective and the objective. At the same time, another important influence of this theory is the idea of context in theory construction. Constructivism explicitly attends to contextual factors in the process of theory creation. For example, a study on environmental dynamics suggests that the actions of the top management team send powerful signals to organizational members and may prompt employees to take action, leading to a more volatile environment [40]. Likewise, the constructivist-based view of knowledge and learning is very different from the traditional view.

From the perspective of knowledge, constructivism is described as a theory of knowledge based on philosophy, psychology, and cybernetics [41]. For a long time, "knowledge," which is a prevalent concept in the development of information systems and other engineering disciplines, has been understood as "verified true beliefs" from a positivist perspective. From this perspective, knowledge is manifested as truth that exists in the form of propositions, which is independent of human perception and understanding and has the characteristics of objectivity, authenticity, and universality. However, from the perspective of constructivism, it is believed that reality is constructed through subjective meaning, shared language and social politics, and this view includes the possibility of multiple realities. Therefore, knowledge is also a construction in a specific context, and it is continuously developed through practice [42].

From the perspective e of learning, constructivism believes that it is meaningless to talk about learning or ability as separate from the real environment, and interaction with the environment is the only way to form ability. Learning occurs in an activity or behavior in a social environment. Learners choose or decide on their own behavior through contact and interaction with the context. Therefore, learning can only be given real meaning if it is embedded in the context in which it is maintained. Lave and Wenger [43] proposed the concept of "legitimate peripheral participation," in which learning refers to the fact that novice learners initially engage in peripheral activities within a community of practice, observe skilled practitioners and participate in activities under their guidance, and gradually develop knowledge and skills in actual work participation. For the community and learners, the deeper aim of these activities is to become a formal member of the community through "participation." This participation is bound to be closely related to context.

Therefore, from the perspective of constructivism, the role of context is emphasized in the learning process. These viewpoints provide theoretical support for establishing the mechanism of absorptive capacity and firm performance and revealing the difference between absorptive capacity and firm performance that arise from context.

## Absorptive capacity and context dependence

When both knowledge and learning are closely related to the specific context of an organization, how does absorptive capacity reflect context-dependent characteristics? Although the existing research on absorptive capacity still focuses on the internal structure of absorptive capacity and the relationship between absorptive capacity and innovation, many studies have proposed that absorptive capacity reflects the characteristics of external contexts to a certain extent and even coevolves with external contexts. Van den Bosch, Volberda [44] emphasized that absorptive capacity is not static but rather changes with the external knowledge environment. In a more chaotic knowledge environment, organizations can achieve broader knowledge acquisition by changing their organizational structure and integration capabilities, thereby improving their absorptive capacity. Therefore, organizational structure and integration capabilities are the main mechanisms through which organizational absorptive capabilities can coevolve with their external knowledge environment. Lewin, Massini [45] expound on the external context and absorptive capacity through the manifestation of absorptive capacity

in organizational routines. Routines, as the basis of capabilities, provide a feedback mechanism for internal and external interactions within the organization, thereby realizing the interaction between the organization and the environment [46]. In addition, existing studies have analyzed many contexts that can moderate absorptive capacity and firm performance, including competitiveness, dynamics, knowledge characteristics, and regime of appropriability. Lichtenthaler [47] has called for a more comprehensive and systematic study of the contextual factors of absorptive capacity.

For context, Cappelli and Sherer [48] define context as external factors that can help explain certain organizational phenomena. The effect of context on research results is subtle and powerful [19]. Context is a significant external environmental feature of an organization, and context may also be an internal implicit factor. According to Johns [19] classification of contexts and the contexts emerging in existing empirical studies on absorptive capacity and firm performance, this paper classifies the contexts of absorptive capacity and firm performance as follows: (1) The research context includes the measurement of absorptive capacity, measurement of firm performance, and types of data. (2) The economic contexts includes the types of countries, types of industries, and level of dynamism faced. (3) The sample context includes the average age and average size of the sample. As shown in Fig 1.

## Theoretical hypothesis

### The moderating effect of the research context

**Moderating effects of absorptive capacity.** On the one hand, when proposing the concept of absorptive capacity, Cohen and Levinthal [8] emphasized the cumulativeness of absorptive capacity, arguing that the concept of absorptive capacity is a function of an organization's existing knowledge base. That is, the accumulation attribute transforms absorptive capacity into a stock concept. When an organization has a greater knowledge base, it can recognize external knowledge, and its innovation and performance can be improved. However, at the same time, this characteristic makes achieving a major strategic adjustment unlikely for the organization. A major result of the cumulativeness of absorptive capacity is path dependence regarding the knowledge base and the expected mechanism. If an organization cannot enter or continue to invest in a certain technology field at an early stage, then it will be locked out of this technical field. On the other hand, some studies elaborate on the dynamic properties of

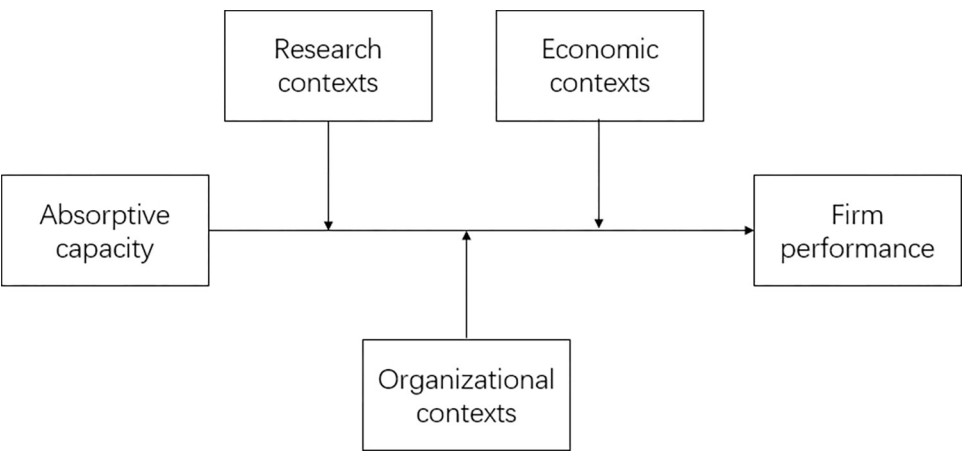

**Fig 1. Research model.**

absorptive capacity. For example, the most representative is the work of Zahra and George [24], which summarized absorptive capacity as a kind of dynamic capacity and proposed that absorptive capacity can rearrange enterprise resources at the strategic level. This dynamic nature enables firms to strategically change and evolve by embedding them into organizational processes and routines. At the same time, absorptive capacity is further divided into potential absorptive capacity and realized absorptive capacity within the process, and potential absorptive capacity, with its functions of identification and assimilation, is considered to be the source of corporate strategic flexibility. The dynamic nature of absorptive capacity enables organizations to adapt more quickly to their external environments and thus to make adjustments and obtain feedback on performance, which is similar to a flow concept.

In fact, Cohen had a similar view when he proposed the concept of absorptive capacity, but the criterion for distinguishing cumulativeness and dynamics is the category of knowledge, that is, whether it is homogeneous knowledge or heterogeneous knowledge. Cohen believes that when there is a large amount of homogenous knowledge within the unit, the absorptive capacity manifests as enhanced communication capabilities, which is more conducive to knowledge cumulativeness. However, when there is a large amount of heterogeneous knowledge within the unit, the absorptive capacity manifests as a stronger change characteristic, which is more conducive to innovation. Thus, differentiated conceptions of absorptive capacity have important implications for how absorptive capacity is measured.

H1a: The means of measurement of absorptive capacity significantly moderates the relationship between absorptive capacity and firm performance.

**Moderating effects of performance.** How to correctly measure firm performance has become an important topic in the field of strategic management in the past decade [49–52]. Currently, scholars have proposed several different standard performance measurement methods. In accordance with the difference between validity and efficiency, performance measurement indicators on the one hand include highly effective indicators such as market satisfaction, corporate reputation, sales volume, market share and new product success rate, while on the other hand, higher efficiency indicators include profit margins, return on assets (ROA), return on equity (ROE) and other indicators. According to different time periods, this indicator is divided into a measurement of current performance and an estimation of future performance. For example, Baker and Sinkula [53] divide enterprise performance into the two categories of effectiveness and adaptability, which are indicators related to the existing success rate and indicators related to actively adapting to environmental changes and seeking opportunities. Venkatraman, N. [54] further divided performance evaluation into financial-related metrics and operational-related metrics.

This paper explores the classification of corporate performance into objective and subjective measures. The objective measurement methods are mainly financial indicators, which are generally obtained through the audited financial statements that are published by enterprises. The subjective measurement method involves the completion of the scale by internal or external personnel, and the subjective measurement method covers financial, operational and commercial indicators. At present, some studies have found that subjective measures of performance are consistent with objective measures of performance [55,56]. However, other studies have found differences in the results of these two measurement methods. For example, some studies on market orientation and firm performance have found that subjectively measured performance has a stronger correlation with performance than objectively measured performance [57]. At the same time, some scholars believe that subjective measurement methods can reveal complex and implicit indicators and dimensions, such as brand equity and

customer satisfaction, to a greater degree than objective measurement methods can. Therefore, the correlation obtained by subjective measurement is higher than that obtained by objective measurement. On the basis of the above discussion, this paper argues the following:

H1b: Performance measurement methods significantly moderate the relationship between absorptive capacity and firm performance.

**The moderating effect of data types.**   Existing studies generally claim that panel data have the following advantages over cross-sectional data for measurement: Panel data can often provide researchers with a large number of samples, increase the statistical degrees of freedom, and reduce the collinearity between explanatory variables, thereby improving the overall validity of the estimates. However, it is difficult to make effective inferences from cross-sectional data in the face of dynamic changes [58]. Therefore, panel data are obviously better able to reflect the temporal characteristics of the samples. Compared with cross-sectional data, panel data can solve the problem of missing variables. Missing variables are often a result of unobservable individual differences or "heterogeneity," and if this heterogeneity does not change over time, then panel data can provide a powerful tool for addressing missing variables, which cross-sectional data cannot provide. Panel data provide dynamic behavioral information on individuals. Panel data have both cross-sectional and time dimensions and can solve problems that cross-sectional data and panel data alone cannot. Therefore, the types of different samples can have a significant impact on the results. Panel data can better reflect cumulative characteristics through time-dependent characteristics, while cross-sectional data can better reflect dynamic characteristics. On this basis, this paper proposes the following:

H1c: Data type significantly moderates the relationship between absorptive capacity and firm performance.

## The moderating effect of economic contexts

**The moderating effect of country type.**   Developed countries generally have a sound intellectual property protection system, and such a sound system has an important impact on the R&D investment of an organization, which in turn has an impact on its absorptive capacity. Enterprises' exclusivity of knowledge includes their ability to effectively protect and benefit from new product research and development [59], while exclusivity depends to some extent on the organization's home country's intellectual property protection system. If patent protection is strong, then enterprises will apply for more patents and, on this basis, generate more complete and more tradable high-quality scientific and technical information [60]. At the national level, developed countries have obvious advantages in knowledge creation and knowledge transformation. At the same time, developed countries have perfect intellectual property protection systems. Therefore, enterprises in developed countries are more willing to invest capital in R&D. In terms of absorptive capacity, there is a clear disparity among different country types [61]. However, enterprises being housed in developing countries can easily lead to disordered competition due to a lack of local knowledge and an intellectual property system. Therefore, absorptive capacity is more likely to be seen as a concern of the external environment and thus left to adjust its own strategies. Therefore, this paper proposes the following:

H2a: Country type significantly moderates the relationship between absorptive capacity and firm performance.

**The moderating effect of environmental dynamics.**   The importance of seeing the external environment is one of the most important contingency factors in organizational

development has been further enhanced in the study of absorptive capacity. First, the external environment directly affects the force with which the absorptive capacity effect is exerted. An increase in environmental dynamics increases the ambiguity of causality and the ability of competitors in the industry to imitate a company's capabilities decline. This limitation enables companies to obtain excess returns and competitive advantages [62,63]. Jansen [64] also found that the benefits of learning to organizational absorptive capacity are proportional to the degree of dynamism in the external environment, with exploitative learning exerting a better effect in a stable environment and a poorer effect in a dynamic environment. Second, knowledge in the external environment affects an organization's future decision-making and capability development [65]. When the environment is more dynamic, an organization tends to explore and discover external knowledge and identify new trends and opportunities. At this time, the organization has a broader vision and a more flexible strategy. When the environment is stable, an organization tends to emphasize improving internal efficiency and promoting the commercialization of internal knowledge. In this context, the organizational strategy lacks flexibility [66]. Additionally, existing research further divides environmental dynamics into technology dynamics and market dynamics [67]. The dynamics of technology include the degree of rapid change and the level of unpredictability of technology. The dynamics of the market can be measured by the degree to which consumer preferences and demands change [68]. Being within a dynamic environment may place some constraints on an organization's existing knowledge base. For example, a dynamic environment not only causes an organization to have a greater need to grasp the preferences of existing customers and increases the likelihood that an organization falls into a capability trap [65,69], but it also destroys the integration of existing knowledge and leads to organizational inertia [69]. In a dynamic environment, continuous updating of the knowledge base and increasing strategic flexibility are the main means of gaining an advantage [63,70]. Therefore, in a dynamic environment, an organization with strong absorptive capacity is better able to gain a competitive advantage and achieve improved corporate performance. For instance, LICHTENTHALER [66] found that when the dynamics of technology increase, firms that acquired external technical knowledge are more likely to achieve better performance levels. In terms of market dynamics, Rerup [71] also pointed out that without absorptive capacity, companies cannot evaluate their product strategies or their success rates. Based on the above discussion, this paper argues the following:

H2b: Environmental dynamics significantly moderate the relationship between absorptive capacity and firm performance.

**Moderating effect of industry type.**   Knowledge is becoming the most important competitive factor for countries and regions. At the same time, research increasingly recognizes the differences in innovation and performance across industries [72]. In particular, knowledge-based or knowledge-intensive industries significantly differ from other industries in their innovation process, knowledge sources, and knowledge coding. Firm output in knowledge-intensive industries depends on a large amount of complex knowledge [73]. In knowledge-intensive industries, the rich knowledge base of enterprises indicates strong absorptive capacity, which in turn makes it easier for enterprises to acquire external knowledge. This process enables companies in high-tech industries to achieve higher levels of innovation and performance. Conversely, in less knowledge-intensive industries, firms tend to gain competitive advantage through capital and scale expansion. This logic ascribes great importance to the impact of industry differences in the relationship between absorptive capacity and firm performance. At the same time, high-tech industries and knowledge-intensive industries are usually used as samples in the literature on absorptive capacity. For instance, to explore the

relationship between absorptive capacity and enterprise innovation, Liao, Fei [74] chose knowledge-intensive industries as their research object. In addition, with the aim of effectively understanding the structure of the absorptive capacity process, Patterson and Ambrosini [33] chose the knowledge-intensive biopharmaceutical industry as a research object.

High-tech industries not only have obvious knowledge-intensive characteristics but also generally exhibit strategic significance for the government, large capital investment, high risks and benefits, and the rapid aging of products and processes, and their levels of technology intensity and complexity are much higher than those of traditional industries. Generally, high-tech industries simultaneously face higher knowledge density and stronger environmental dynamics. On the whole, it is the combination of knowledge accumulation and environmental dynamics that is important to high-tech industries. On the other hand, in some articles on the relationship between absorptive capacity and firm performance, industry attributes are often used as control variables or moderator variables. Therefore, this paper uses industry attributes as moderating variables. On the basis of the above discussion, this paper proposes the following:

H2c: Industry type significantly moderates the relationship between absorptive capacity and firm performance.

**Moderating effects of organizational context.** Volberda, Foss [75] argue that firm size is an important source of absorptive capacity heterogeneity. A basic theoretical source for this position is Schumpeter's assertion that innovation activities are mainly provided by large-scale firms [76]. Large enterprises can invest greater R&D resources, so large-scale enterprises have significantly higher levels of knowledge accumulation than small-scale enterprises [77,78]. On the other hand, small-scale enterprises are more inclined to actively search out, utilize and integrate external knowledge [79]. At the same time, small-scale enterprises are less affected by bureaucracy, so coordination is easier for them.

In terms of enterprise years, some researchers believe that mature enterprises have good absorptive capacity because they have accumulated much experience and transformed that experience into conventions or institutions. Mature enterprises have a better developed reputation and social status, which makes it easier for enterprises to enter the existing organizational network and thus more easily obtain external knowledge. In addition, mature firms are more likely to have better human resources for the discovery and effective use of external knowledge. On the other hand, although new ventures have higher innovation capabilities [80] and are not easily affected by organizational inertia, they can quickly respond to external knowledge. However, start-ups generally lack the ability to commercialize knowledge, which is an advantage of more established companies. In light of the above inconsistency, it can be found that the age and scale of an enterprise are also important moderating factors. On the basis of the above discussion, this paper proposes the following:

H3a: Firm size significantly moderates the relationship between absorptive capacity and firm performance.

H3b: Firm age significantly moderates the relationship between absorptive capacity and firm performance.

## Research methods

Meta-analysis, a quantitative synthesis method for reanalyzing existing empirical studies, is currently a common international aggregation technique. There is a growing body of empirical research on absorptive capacity and organizational performance, but the findings of this

research are somewhat contradictory. Meta-analysis can unify independent studies and overcome the limitations of using a single sample, and at the same time, it can weaken the errors caused by the research design context and the subjective bias of the researcher and present the relationship between variables in a large sample in a more realistic way so that the research conclusions are more accurate.

## Sample

To collect the data needed for this study, we searched various databases, including EBSCO, Web of Science, Elsevier ScienceDirect, SpringerLink, Wiley Online Library, and ProQuest. To ensure the quality and quantity of the acquired samples, the main source of data acquisition for this paper was the Web of Science core collection, while other databases were also searched separately to check for missing items. To obtain the most effective search results, this paper used the search formula "(absorptive capacity OR absorptive capacities OR ACAP OR ACAP capability) AND (performance OR firm performance OR financial performance OR organizational performance)" as the search content. In terms of time, this paper started with the first introduction of absorptive capacity in 1990 and used 1990–2021 as the search interval. A total of 6730 search results were obtained.

Among the above search results, we found that some of the studies mentioned absorptive capacity or organizational performance but used a research topic that was not related to absorptive capacity. Therefore, to further refine the research topic and scope, the authors of this paper read the abstracts of the above literature to determine whether the literature used absorptive capacity and organizational performance as the article topic. Finally, the relevant literature was narrowed to 301 articles.

To conduct a META analysis, the corresponding studies had to possess the following characteristics: the study had to be empirical and contain parameters that could be translated into study effect sizes, including correlation coefficients, regression coefficients, or path coefficients. Therefore, after excluding theoretical studies, case studies, and modeling and simulation studies from the sample, the total number of studies that met all criteria was 58 (Fig 2).

## Coding

In this paper, a coding manual was constructed to collect various information needed for the META analysis, including author, journal, year of publication, effect size, and moderating variables. Based on the contents of the coding manual, the coding procedure for the literature was conducted independently by two researchers. The coded results were checked by the two researchers according to the recommendations of Schmidt and Hunter [81], and any inconsistent coding content was discussed to achieve a consistent coding result.

The moderating variables were coded as follows: in terms of research methods, absorptive capacity as measured by R&D was coded as cumulative, while absorptive capacity as measured by the main portrayal process in the scale was coded as dynamic. In the measurement of business performance, objective data-based indicators such as ROA, ROI, ROS, ROE, Tobin's Q, and sales growth were coded as objective performance measurements, while scales were coded as subjective performance measurements. Among the data types, to examine the effect of time, the data types in this paper were coded as cross-sectional data and panel data. In terms of contextual factors, this paper coded the sample into developed and developing countries based on the study's country of origin. On the basis of industry classification criteria, the type of industries studied were coded in the sample as high-technology and nonhigh-technology industries. Based on the definition of environmental dynamism of [82], the sample literature is coded in this paper as utilizing either dynamic or nondynamic environments. Finally, regarding firm

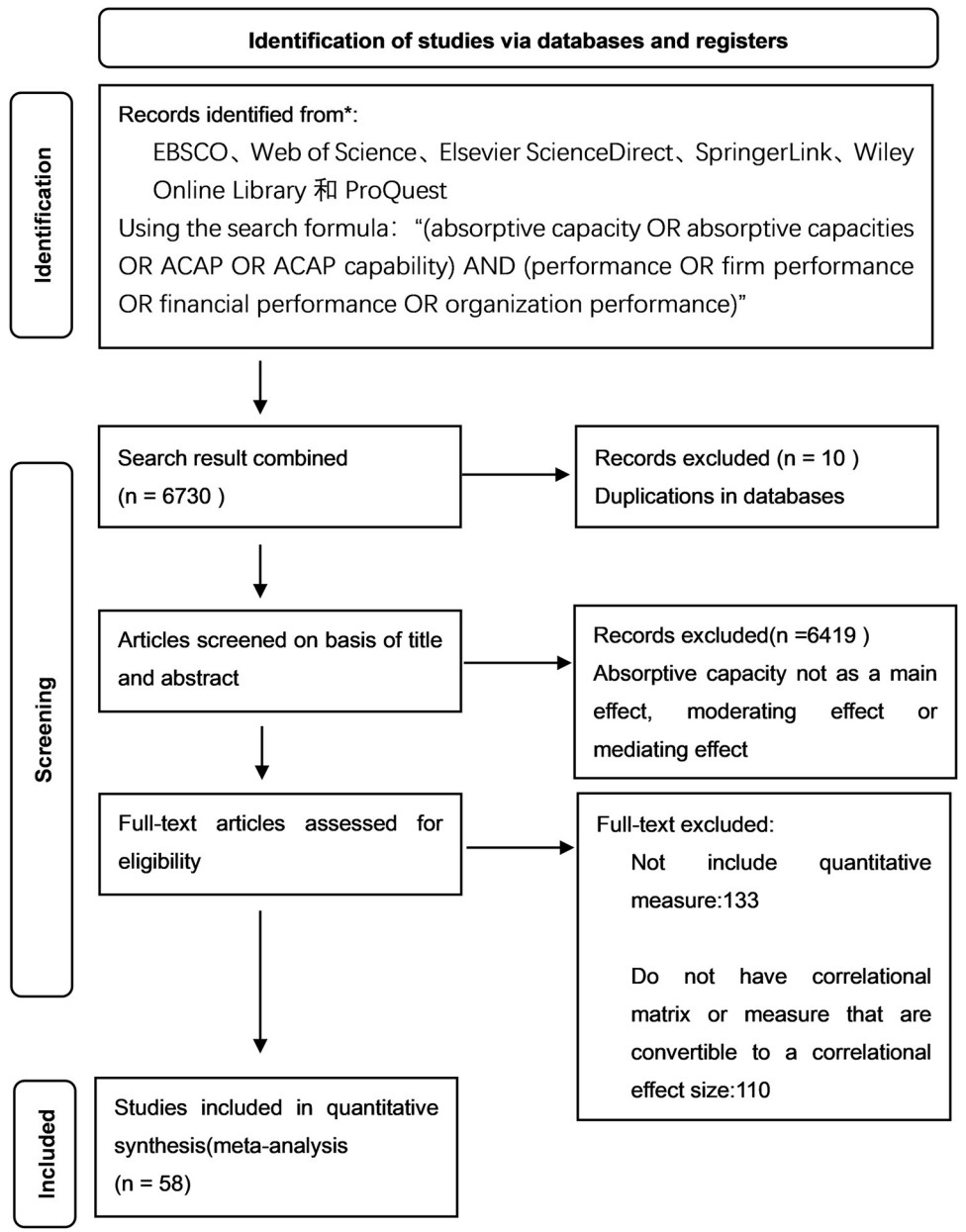

**Fig 2. Sample search process.**

size and age, this paper coded the mean of firm size and age in the sample as the values of the moderating variables.

## Meta-analysis procedures

The meta-analysis procedure suggested by [81] was used in this study. First, to calibrate the reliability of the measurement, Cronbach's alpha coefficient was used as a calibration in reporting the correlation coefficient. The true correlation coefficient $\rho_i$ is calculated as $\rho_i = \frac{r_i}{\sqrt{\alpha_{xi}\alpha_{yi}}}$, where $r_i$ is the sample correlation coefficient of the $i$th study without measurement error correction, and $\alpha_{xi}$ and $\alpha_{yi}$ are the $\alpha$ confidence coefficients of the independent variable x

and the dependent variable y of the $i$th study, respectively. When Cronbach's alpha was provided in the original literature, the correlation coefficient divided by Cronbach's alpha was used as the adjusted effect size for input, and when Cronbach's alpha was not provided in the literature, the mean was used as an adjustment. Subsequently, all adjusted correlation coefficients $\rho_i$ were input as effect sizes in this paper, while Comprehensive Meta-Analysis(CMA 3.0) was chosen as the analysis software, and $\bar{\rho} = \frac{\sum_{i=1}^{k} W_i \rho_i}{\sum_{i=1}^{k} W_i}$, where $W_i$ is the sample size of the $i$th study and $\rho_i$ is the corrected correlation coefficient of the ith study, was used as the formula for calculating the combined effect size. Finally, the Q-statistic values were calculated on the basis of the combined effect values to test for heterogeneity in the sample.

In this paper, the categorical moderating variables are examined using meta-subgroup analysis. Subgroup analysis is one of the common techniques used in meta-analysis to analyze the entire set of data into different subgroups according to the study or individual characteristics and is used to explore the sources of meta-analysis heterogeneity or to answer specific questions related to particular samples, types of interventions, types of study designs, etc. It also allows for the comparison of different subgroups and effect sizes among these variables. Using Meta-subgroup analysis is very similar to the use of t tests and ANOVA in that it analyzes the variance components of the effect sizes of the groups to determine whether the means of the different groups are significantly different. Meta-subgroup analysis uses tests that apply categorical variables. Therefore, in this paper, categorical variables such as situational factors are used as subgroup factors to find the moderating effect and strength of situational factors on size of the effects.

In this paper, meta-regression analysis is used to test continuous moderating variables. Meta-regression refers to the use of regression in META analysis to analyze the effect of one or more moderating variables on the effect values. Unlike subgroup analysis, meta-regression can be used to analyze continuous moderating variables. In this paper, we use meta-regression to analyze the impact of firm size and firm age on effect size. The regression equation is as follows:

$$\rho_i = \beta_0 + \beta_1 x_{1i} + \beta_2 x_{2i} + \cdots + \zeta_i + \varepsilon_i$$

where $\rho_i$ is the effect size in the ith study, $\beta_0$ is the mean effect size after controlling for other study characteristics, $x_{1i}$, $x_{2i}$, etc., are the different moderating variables of the study, $\beta_1$, $\beta_2$, etc., are the regression coefficients of the different moderating variables, $\zeta_i$ is the between-study sampling error, and $\varepsilon_i$ is the within-study sampling error.

## Publication bias test

The most common form of publication bias refers to statistically significant findings being more likely to be published by academic journals than nonsignificant findings. Meta-analysis requires a test for publication bias to determine the confidence level in the estimates of the effect sizes within the published literature [83,84].

Funnel plots are currently one of the most important means of testing for publication bias. The funnel plot shows the relationship between the data effect value Fisher's Z and its standard error. If there is no publication bias, then the effect values are randomly and symmetrically distributed around the mean value. If there is publication bias, then the distribution is clearly asymmetric and tends to be in the same direction. The funnel plot of this study is shown in Fig 3. Most of the studies are in the upper part of the funnel plot and are clearly symmetric, while some of the small sample studies have large sampling errors and are therefore distributed over a wider range of values, which indicates that there is no publication bias present.

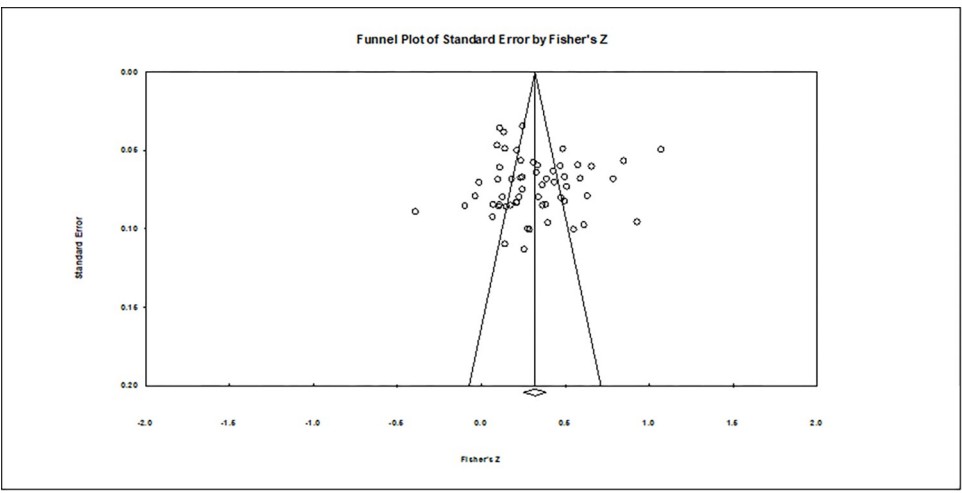

**Fig 3. Funnel chart.**

Next, because of the subjectivity of funnel plots in estimating publication bias, Rosenthal and Robert [85] proposed the concept of fail-safe N, which is a method of calculating how many study documents need to be omitted for a study finding to be statistically nonsignificant. In this paper, the fail-safe number is 9186, which means that 9186 documents that were not included in the meta-analysis need to be omitted to make it statistically nonsignificant. Finally, based on the quantitative regression analysis proposed by Egger and Smith [86] for the presence of publication bias, the test results show a T value of 0.229 and a p value of 0.819 for the two-sided test, which are both much greater than the significance level of 0.05, indicating that there is no publication bias present. Thus, there is no publication bias in this study.

## Results of meta-analysis

To explore the relationship between absorptive capacity and firm performance, this paper obtains a total of 58 effect sizes and a total of 13,783 observations in the research sample. Table 1 describes the statistical characteristics of the sample in this study, including the overall effect sizes under different models and their confidence intervals [87]. In the meta-analysis, the confidence interval indicates the possible interval of the overall effect size. A confidence interval that does not include 0 indicates that the relationship is statistically significant.

### Heterogeneity test

Heterogeneity testing is a critical step in meta-analysis. On the one hand, it is necessary to determine whether to use a fixed-effects model or a random-effects model to analyze the data based on the heterogeneity judgment before performing a meta-analysis and meta-regression on the literature. On the other hand, when the results show that the sample is heterogeneous, the data in the literature may come from different populations, and further analysis of the

**Table 1. Heterogeneity test results.**

| model | K | Effect size | 95% confidence interval | | Two-tailed test | | Heterogeneity test | | | |
|---|---|---|---|---|---|---|---|---|---|---|
| | | | Lower limit | Upper limit | Z value | P value | Q value | Degree of freedom | P value | I-squared |
| **Fixed effect model** | 58 | 0.309 | 0.293 | 0.324 | 36.933 | 0.000 | 862.36 | 57 | 0.000 | 93.39 |
| **Random effect model** | 58 | 0.309 | 0.247 | 0.368 | 9.370 | 0.000 | | | | |

differences between the populations or the reason for the differences is needed. The test results of this paper are shown in Table 1.

According to the test results, the Q value of the study was 862.36, the df(Q) was 57, and the P value was less than 0.001, indicating that there was obvious heterogeneity among the literature. Meanwhile, the I-squared value was 93.39, indicating that 93.39% of the heterogeneity was caused by differences between studies, and 6.61% of the heterogeneity was caused by random errors. The results showed that there was heterogeneity between studies, so a random-effects model should be used for meta-analysis and meta-regression. In addition, further discussion is needed to identify the sources of heterogeneity, that is, the potential moderating variables in the relationship between absorptive capacity and firm performance that may cause differences in effect sizes.

## Meta subgroup analysis

On the basis of heterogeneity among studies, this paper divides the research literature into different subgroups according to the assumption of moderating variables and conducts heterogeneity analysis on different subgroups to determine whether the heterogeneity between groups is significant.

In measuring absorptive capacity, the measurement of absorptive capacity is divided into cumulative and dynamic aspects according to theoretical assumptions. Through meta-subgroup analysis, it was found that the effect size of absorptive capacity and firm performance as based on dynamicity was 0.326 (P<0.05), while the effect size of absorptive capacity and firm performance as based on accumulation was 0.194 (P<0.05). The differences between groups were statistically significant, which shows that Hypothesis H1a is verified; that is, the measurement of absorptive capacity significantly impacts the effect of absorptive capacity on firm performance. When a firm's absorptive capacity measurement is cumulative, it has a lower performance level, and when that absorptive capacity measurement is dynamic, a firm has a higher performance level. Absorptive capacity that appears to be cumulative requires a longer period of time to affect performance.

In terms of performance measurement and data types, this paper divides performance measurement methods into subjective measurement methods and objective measurement methods. Through a between-group analysis, it was found that the overall effect value of the objective measurement of performance in the mixed-effects analysis was 0.138 (P<0.05) (as shown in Table 2), while the overall effect value of the subjective measurement of performance was 0.4 (P<0.05), which was significantly higher than the objectively measured effect value, but at the same time, the Q value of the difference between groups was 25.327 (P<0.05). Thus, H1b is validated. This shows that the measurement of subjective performance conjoins both psychological bias and measurement integrity, making subjectively measured performance levels higher than objectively measured performance levels. For panel data, the effect value is 0.146 (P<0.05), and the effect value for cross-sectional data is 0.323 (P<0.05). The effect value obtained from cross-sectional data is higher than that obtained from panel data. The difference is significant, confirming H1c and indicating that after adding the effect of time, panel data can better reflect the accumulation of absorptive capacity, resulting in a lower effect size.

In testing economic context, this paper found that the effect size of developed countries was 0.255 (P<0.05), while the effect size of developing countries was 0.376 (P<0.05), and the difference was statistically significant (P<0.05). Thus, H2a is verified. This indicates that developed countries have generally higher knowledge levels and, at the same time, a sound intellectual property system, which makes enterprises more inclined to pursue knowledge accumulation and R&D investment, resulting in more recordable costs. Developing countries,

**Table 2. Meta-subgroup test results.**

| Variable | Studies, K | Sample size, N | Effect size | 95%CI | | Z value | Heterogeneity test | | |
|---|---|---|---|---|---|---|---|---|---|
| | | | | Lower limit | Upper limit | | df | Q value | P value |
| **Overall** | 58 | 13783 | 0.309 | 0.247 | 0.368 | 9.37 | | | 0.000 |
| **ACAP measurement** | | | | | | | | | |
| Dynamic | 50 | 11792 | 0.326 | 0.257 | 0.392 | 8.757 | | | 0.000 |
| Cumulative | 8 | 1991 | 0.194 | 0.119 | 0.268 | 4.992 | | | 0.000 |
| Between | | | | | | | 1 | 6.548 | 0.010 |
| **Performance measurement** | | | | | | | | | |
| Objective | 20 | 5695 | 0.138 | 0.07 | 0.204 | 3.95 | | | 0.000 |
| Subjective | 38 | 8088 | 0.391 | 0.32 | 0.458 | 9.876 | | | 0.000 |
| Between | | | | | | | 1 | 25.327 | 0.000 |
| **Data type** | | | | | | | | | |
| Panel | 5 | 793 | 0.146 | 0.043 | 0.246 | 2.769 | | | 0.006 |
| Cross section | 53 | 12990 | 0.323 | 0.258 | 0.384 | 9.287 | | | 0.000 |
| Between | | | | | | | 1 | 8.549 | 0.003 |
| **Country type** | | | | | | | | | |
| Developed country | 33 | 7613 | 0.255 | 0.178 | 0.329 | 6.293 | | | 0.000 |
| Developing country | 25 | 6170 | 0.376 | 0.281 | 0.464 | 7.258 | | | 0.000 |
| Between | | | | | | | 1 | 3.879 | 0.049 |
| **Industry type** | | | | | | | | | |
| High-tech | 18 | 3482 | 0.279 | 0.116 | 0.428 | 3.301 | | | 0.001 |
| Non high-tech | 40 | 10301 | 0.32 | 0.26 | 0.378 | 9.93 | | | 0.000 |
| Between | | | | | | | 1 | 0.234 | 0.628 |
| **Environmental dynamics** | | | | | | | | | |
| Dynamic | 15 | 2995 | 0.141 | 0.042 | 0.237 | 2.775 | | | 0.006 |
| Nondynamic | 43 | 10788 | 0.363 | 0.294 | 0.428 | 9.661 | | | 0.000 |
| Between | | | | | | | 1 | 13.621 | 0.000 |

on the other hand, put more emphasis on knowledge application and response to environmental changes and have better performance levels over the short term. In terms of the impact of environmental dynamics, when faced with a dynamic environment, the effect size of absorptive capacity and firm performance (0.141) is significantly lower than that under a nondynamic environment (0.363) (H2b), indicating that although individual cases or specific sample studies show that absorptive capacity is beneficial to coping with a dynamic environment, from the perspective of a large sample, a dynamic environment increases the cost of accumulating and acquiring knowledge and reduces performance levels. For industry types, there is no statistically significant difference between the effect size of high-tech industries (0.279) and that of nonhigh-tech industries (0.32). Thus, H2c is not verified. Industry factors may impart a more complex effect on the relationship between absorptive capacity and firm performance because high-tech industries have both high knowledge content and highly dynamic environmental characteristics, making the role of absorptive capacity more contingent with more complex feedback mechanisms.

## Meta regression analysis

To observe continuous variables such as firm scale and firm age, this paper uses meta-regression for verification. Table 3 shows the results of the meta-regression on firm size and age. Model 1 is a basic model using measurement factors and situational factors as control

**Table 3. Results of meta-regression analysis.**

| Moderator variable | Model 1 | | | Model 2 | | | Model 3 | | |
|---|---|---|---|---|---|---|---|---|---|
| | coefficient | S.E | p | coefficient | S.E | p | coefficient | S.E | p |
| Intercept | 0.0016 | 0.0825 | 0.9843 | 0.1353 | 0.1071 | 0.2061 | 0.1269 | 0.1088 | 0.2434 |
| Performance measurement | 0.2076 | 0.0936 | 0.0265 | 0.1891 | 0.0866 | 0.0289 | 0.1833 | 0.0875 | 0.0361 |
| Data type | 0.1351 | 0.1387 | 0.3301 | 0.17 | 0.1294 | 0.1891 | 0.1605 | 0.131 | 0.2207 |
| Country type | -0.1173 | 0.1154 | 0.3093 | -0.1386 | 0.1066 | 0.1935 | -0.109 | 0.1315 | 0.4068 |
| Industry type | 0.0676 | 0.0976 | 0.4881 | 0.142 | 0.099 | 0.1514 | 0.121 | 0.1127 | 0.2828 |
| Environment dynamic | 0.2125 | 0.1123 | 0.0584 | 0.1666 | 0.1062 | 0.1168 | 0.1386 | 0.1288 | 0.2817 |
| Firm size | | | | -0.0488 | 0.0272 | 0.0727 | -0.0462 | 0.0279 | 0.0974 |
| Firm age | | | | | | | 0.0017 | 0.0043 | 0.7028 |
| $R^2$ analog | $R^2 = 0.48$ | | | $R^2 = 0.58$ | | | $R^2 = 0.59$ | | |

variables. Model 2 adds firm size on the basis of control variables and finds that firm size has a negative moderating effect on absorptive capacity and firm performance (H3a). This demonstrates that with an increase in the size of an organization, the accumulative effect of the absorptive capacity is strengthened; However, the dynamic of the absorptive capacity is weakened, and the enterprise cannot quickly adjust its strategic state to adapt to these environmental changes. Model 3 adds organizational age on this basis and finds that the effect size of a company's age on its absorptive capacity and corporate performance is limited, indicating that an increase in a company's age will not have a significant impact on absorptive capacity and corporate performance. Alternatively, the impact of firm age on the effect size is offsetting; that is, as firm age increases, the effects of knowledge growth and firm inertia may cancel each other out so that firm age does not have a significant impact on effect size (H3b).

# Discussion

## Main conclusion

The rapid growth of research on absorptive capacity reflects the importance of absorptive capacity theory. However, in the literature review, two problems with absorptive capacity theory are identified: the abstract nature of the absorptive capacity concept and the variability of the empirical results on absorptive capacity. The abstraction problem stems from the definition of absorptive capacity itself, which is the result of an analogy between corporate knowledge acquisition processes and computer information processing processes that is based on cognitivist learning theory. The variability of the empirical results, on the other hand, stems from the neglect of specific contexts and the lack of a systematic assessment of absorptive capacity and firm performance. At the same time, the issues of abstraction and variability are interconnected and mutually reinforcing. Therefore, to address these issues, this paper first proposes the source of abstractness in the concept of absorptive capacity through a theoretical review and introduces a constructivist perspective on knowledge and learning to establish the link between absorptive capacity and enterprise practices. At the same time, it is argued that the degree of involvement of enterprises in knowledge acquisition practices is equally important as the knowledge base.

At the same time, significant between-study differences, other than sampling errors, were found based on the meta-heterogeneity test. To further determine the sources and influencing factors of these differences, meta-subgroup tests and meta-regression analyses were conducted on the samples used in this study. Starting from the measurement factors of absorptive capacity, the cumulative effect of absorptive capacity on effect size is significantly lower than the

effect size of absorptive capacity dynamics. This result validates the hypothesis of existing studies regarding the concept of the multidimensionality of absorptive capacity and illustrates the impact of measurement factors on absorptive capacity. Absorptive capacity that exhibits accumulation has a lesser impact on current firm performance, while absorptive capacity that is dynamic has a greater impact on current firm performance. In terms of measurement factors, this paper corroborates that the measurement method for firm performance has a significant impact on effect sizes and that the method of performance measurement may impact effect sizes through cognitive biases and measurement differences. Regarding the type of data, panel data better respond to the impact of the cumulative nature of absorptive capacity due to its temporal characteristics. In contrast, cross-sectional data better reflects the dynamic characteristics of absorptive capacity.

In terms of economic context, developed countries exhibit the cumulative nature of absorptive capacity due to better knowledge stock and intellectual property systems, while developing countries mainly exhibit the dynamic nature of absorptive capacity due to the backwardness of their knowledge systems and their lack of institutions. In terms of environmental dynamism, additional costs may be incurred to reduce firm performance due to rapid environmental changes, while non-dynamism engenders a higher level of performance due to the role of knowledge accumulation. In terms of industry influences, the differences are nonsignificant due to the role of dynamism and the complexity of the knowledge base in high-technology industries. From the characteristics of the sample scenario, it can be seen that the combination of decreased ability of a firm to regulate changes in strategy and the increased costs resulting from the increase in size, as well as the accumulation characteristics of absorptive capacity, influences the level of negative regulation on firm size. Likewise, firm age is more complex, and thus, its moderating effect is nonsignificant.

## Implications for research and practice

The contribution of this paper to the literature is as follows. First, despite the rapid growth of research on absorptive capacity, the findings do not well explain the differences in empirical results between absorptive capacity and firm performance or their mechanisms [88]. Many scholars have begun to notice the effect problem in the empirical studies on absorptive capacity, and the effect problem may stem from the ambiguity in the definition of absorptive capacity itself [18,89]. This paper reveals the variability of the results of existing empirical studies by classifying the measures of absorptive capacity as cumulative and dynamic in a meta-analysis. The results support the literature on absorptive capacity, suggesting that the relationship between absorptive capacity and firm performance is largely influenced by the construct measurement factors, and the extent of this influence is profound. The portion of the studies that treat absorptive capacity as a stock obtain results that are more prone to path-dependent properties and indicate lower levels of performance [31,90]. Studies that consider absorptive capacity as a set of practices that can be changed, or as having dynamic capacity, indicate better performance levels [5,24,91]. The results from the META analysis show that this difference in perspective still exists in the study of absorptive capacity. Likewise, the same problem exists in the measurement of performance and the type of data used, with researchers showing preference to a particular scale or research method that facilitates the results. All these issues can further contribute to the differentiation of the concept of absorptive capacity. Therefore, further refinement of the concept of absorptive capacity still needs to be enacted.

Second, the literature agrees that absorptive capacity is an intrafirm determined capability, i.e., the strength of absorptive capacity is determined by internal organizational structure, knowledge networks, and other factors [92]. However, few studies have been conducted to

summarize the external contexts that specifically address the relationship between absorptive capacity and firm performance. Contextual factors provide a very important basis for knowledge acquisition and transfer; for example, contexts with well-developed systems are more conducive for the establishment of trust between firms and can effectively facilitate knowledge transfer [93,94]. Similarly, for absorptive capacity, the external environment provides the basis from which a firm's absorptive capacity can function. Therefore, this paper provides a test of absorptive capacity in different specific economic contexts and draws conclusions.

Finally, organizational size and age have been key issues in extant absorptive capacity studies [95]. However, Volberda, Foss [96] note that firm size is a key source of heterogeneity for ACAP, and the lack of research on firm size is surprising. The current findings on firm size and absorptive capacity are unclear. However, it is generally accepted that large firms have more resources to invest in R&D [78,97], so they have a stronger absorptive capacity than small firms. For the age of the business, mature companies have a better reputation for acquiring external knowledge [98]. At the same time, established companies are better able to embed broader knowledge networks to access knowledge. Younger firms, on the other hand, may have a higher motivation to acquire knowledge, and knowledge acquisition is simpler for younger firms due to fewer practices and inertia. The findings of this paper are similar to those in the literature. The effects of organizational size and age on firm absorptive capacity are mixed. The moderating effect on organization size and age needs to be assessed under more stringent conditions.

## Limitations and future research

Although this paper overcomes the problem of research bias inherent in previous individual studies and obtains a more comprehensive research result, it still has some shortcomings. First, the existing empirical studies utilize different understandings of the variables, and due to the need for coding, this paper conducts a conceptual simplification and merging of the effect variables. Although this simplification has theoretical and data support and does not affect the validity of the study, it causes some information loss and potentially biases the conclusion. Second, due to the characteristics of META analysis, factors such as country type and industry type are chosen as moderating variables in this paper. These variables lack novelty to some extent. Therefore, the META analysis may provide a limited theoretical contribution. Finally, the main relationship tested in this paper is that between absorptive capacity and firm financial performance, and thus, there may be a research gap in the study.

To cope with the above limitations, future research should further explore the following aspects. Since the concept of absorptive capacity is still ambiguous, the question of what exactly absorptive capacity is needs to be theoretically explored. Second, the relationship between absorptive capacity and firm performance deserves further exploration. For example, the mechanism by which absorptive capacity acts on firm performance is unclear, and whether there are other mediating or mediating variables that need to be explored by the study. Finally, the social attribute aspect of absorptive capacity is another issue that should be studied, as a large number of studies have focused on the natural attributes of absorptive capacity, such as type of knowledge and organizational structure, while neglecting the impact of the social attributes of people on absorptive capacity.

## Supporting information

**S1 Appendix. Appendix.**
(PDF)

**S2 Appendix. PRISMA_2020_flow_diagram_new_SRs_v1(1).**
(DOCX)

**S3 Appendix. PRISMA_2020_checklist.**
(DOCX)

**S4 Appendix. PRISMA_2020_abstract_checklist.**
(DOCX)

**S5 Appendix. Include list.**
(DOCX)

**S6 Appendix. Minimal data set.**
(XLSX)

## Author Contributions

**Conceptualization:** Kun Pu.

**Data curation:** Kun Pu.

**Formal analysis:** Kun Pu.

**Investigation:** Kun Pu.

**Supervision:** Wei Liu.

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
