## [Decision Letter · Decision Letter 0]

6 Dec 2022

PONE-D-22-28180Is absorptive capacity the "panacea" for organization development? META Analysis of Absorptive Capacity and Firm Performance from the Perspective of ConstructivismPLOS ONE

Dear Dr. Pu,

Thank you for submitting your manuscript to PLOS ONE. After careful consideration, we feel that it has merit but does not fully meet PLOS ONE’s publication criteria as it currently stands. Therefore, we invite you to submit a revised version of the manuscript that addresses the points raised during the review process. We now have three review reports on your manuscript. As you can see, two reviewers have recommended Major Revisions whereas Reviewer 3 have suggested rejection. Overall, in light of the comments received, I have decided to assign Major Revision decision to your manuscript. It is a borderline decision therefore, I suggest you to kindly read through the comments carefully, incorporate them and resubmit your paper.

We look forward to receiving your revised manuscript.

Kind regards,

Muhammad Ali, PhD

Academic Editor

PLOS ONE

Journal Requirements:

Additional Editor Comments:

Dear Author

We now have three review reports on your manuscript. As you can see, two reviewers have recommended Major Revisions whereas Reviewer 3 have suggested rejection. Overall, in light of the comments received, I have decided to assign Major Revision decision to your manuscript. It is a borderline decision therefore, I suggest you to kindly read through the comments carefully, incorporate them and resubmit your paper.

Sincerely

Academic Editor

Reviewers' comments:

Reviewer's Responses to Questions

**Comments to the Author**

1. Is the manuscript technically sound, and do the data support the conclusions?

Reviewer #1: Yes

Reviewer #2: Yes

Reviewer #3: Partly

2. Has the statistical analysis been performed appropriately and rigorously? 

Reviewer #1: Yes

Reviewer #2: Yes

Reviewer #3: Yes

3. Have the authors made all data underlying the findings in their manuscript fully available?

Reviewer #1: Yes

Reviewer #2: Yes

Reviewer #3: No

4. Is the manuscript presented in an intelligible fashion and written in standard English?

Reviewer #1: Yes

Reviewer #2: No

Reviewer #3: Yes

5. Review Comments to the Author

Reviewer #1: Thank you for the opportunity to review this paper. Overall, this is a good paper. However, I have suggestions that can enhance the quality of this work. The authors may consider these suggestions to improve the paper. I wish the authors all the best with the revision.

This paper first introduces the constructivist view of knowledge and learning, discusses the shaping of knowledge and learning by contextual factors, and further explores the influence of context on absorptive capacity. It also performs a meta-analysis.

The introduction of the paper is well-written and it convincingly motivates the case for this study and explains why the study is needed.

The authors have covered much of the literature. However, some recent relevant studies have been overlooked. I provide below some recent studies that are relevant to the study. I suggest the authors look at the recent works of Naqushbandi and Jaisumudin the last 5 years to identify research that strengthens the arguments presented in this paper. Their works on absorptive capacity, knowledge management and organizational context can help the authors develop the theoretical foundation of the paper. The authors may include appropriately in their literature review and discussion the following recent studies:

#The linkage between open innovation, absorptive capacity and managerial ties: A cross-country perspective. Journal of Innovation & Knowledge, (2022). 7, 100167.

#Knowledge infrastructure capability, absorptive capacity and inbound open innovation: evidence from SMEs in France. Production Planning & Control, (2019). 30, 893-906.

#The interplay of leadership, absorptive capacity, and organizational learning culture in open innovation: Testing a moderated mediation model. Technological Forecasting and Social Change, (2018). 133, 156-167.

#Intervening role of realized absorptive capacity in organizational culture–open innovation relationship: Evidence from an emerging market. Journal of General Management, (2017). 42, 5-20.

#Managerial ties and open innovation: examining the role of absorptive capacity. Management Decision, (2016). 54, 2256-2276.

Typo on page 20 L 2 "3. theoretical hypothesis". Also, I suggest the authors remove the word "measurement throughout. Eg 3.1.1 Moderating effects of absorptive capacity measurements should be "3.1.1 Moderating effects of absorptive capacity" Same for Section 3.1.2

Some of the tables (eg table 6) cannot be fully read due to improper formatting.

The authors have done a good job at explaining the study's findings. However, relating the findings to the extant literature - which is the hallmark of a good discussion - is missing. Please update the discussion section to reflect how your findings are in line or in contrast with previous studies.

The authors should highlight the implications of the paper for theory and practice. This should be done in detail to highlight the contribution of the paper.

Some language-related errors and typos exist – please have the paper proofread/edited.

I see potential in this paper. The authors need to revise it based on the feedback above. I wish the authors all the best and look forward to reading an improved version of the study. All the best.

Reviewer #2: This paper presents a qualitative theoretical discussion and a quantitative META analysis to examine the reasons for the apparent discrepancy between the empirical results of absorptive capacity and firm performance in existing studies. In the process of theoretical discussion, this paper finds that the core concept of absorptive capacity is based on cognitivism, and the process of absorptive capacity is based on mechanical cognitive information processing, which focuses on the internal cognitive structure and process of the subject. However, due to the dynamic and complex nature of social phenomena, the process model cannot effectively respond to the influence of contextual factors on their relationships. Therefore, this paper first introduces the constructivist view of knowledge and learning, discusses the shaping of knowledge and learning by contextual factors, and further explores the influence of context on absorptive capacity and constructs a model of the role of context on absorptive capacity and firm performance. Secondly, the paper analyzes the results of the existing empirical studies by META analysis and examines different contexts, and finds that the role of absorptive capacity on firm performance has obvious contextual characteristics, among which: country context, industry context, dynamic context, and firm size have significant moderating effects on absorptive capacity and firm performance. In addition, the research context also has an important impact on the relationship, including the way absorptive capacity and firm performance are measured and the type of data. In summary, this paper expands the meaning of absorptive capacity and examines the role of context in moderating absorptive capacity and firm performance. The abstract is too long. It should focus on the aim, the methods and the results. The methodology should further explained. Finally, the text should be English proofread because some sentences are not clear.

Reviewer #3: Absorptive capacity is an established construct that is frequently used as a moderator of the effect of external resource involvement on firm performance. The model chosen for the study does not examine this and examines the direct correlation with performance. This does not correspond to the definitional core of the theory. At the same time, this creates ambiguity of differentiation of AC from related management constructs such as innovativeness. Why was the approach chosen?

The relevance of the constructivism perspective to the study is not comprehensible. I recommend deleting this paragraph.

The focused moderators are sound. However, they are of limited conceptual value. It would be interesting to see other contextual variables, such as the entrepreneurial objective (e.g., exploration, exploitation), the type of external collaborators involved (e.g., firms, universities), and the form of collaboration....

The differences in the measurement of Absorptive Capacity are not comprehensible. Please, specify the two types of measurement investigated. What exactly was measured in the underlying studies? Currently, the exact definition of AC and thus the message of the meta-analysis is unclear. Alternatively, I suggest distinguishing unidimensional measurements /e.g. R&D intensity), multidimensional measurements, and subjective measurements in the analysis.

The measurement of performance only considers differences between subjective and objective measurements. More relevant would have been contextual distinctions between financial performance and technological performance, and between long-term and short-term performance.

The inclusion criteria of the studies are unclear. If I apply the search string (with the correct brackets) more than 6000 studies are included in the results only in web of science. Why has your search only identified 316 studies? Were only studies that measured financial performance included? Why no studies examining R&D performance? Some studies such as Lane et al. (2001), Flatten et al. (2011), Tzokas et al. (2015), or Melnychuk et. al. (2021) - just to name a few - are not included. Why not? The authors should provide a complete list of studies taken into account and not taken into account.

Some meta-analyses exist that examine absorptive capacity in the context of innovation, strategic alliances, and supply chains. How does the authors' meta-analysis distinguish itself from these?

Overall, the field of absorptive capacity is very established in the management literature. There are also some systematic reviews and meta-analyses available. In the current version of the study, it is not clear how the authors distinguish themselves from it. This is also evident in the very short discussion, which hardly considers conceptual implications from the analysis. What can practice and science learn from the analysis? PLOSONE is an interdisziplinary journal. As such this implication should also help other disciplines.

6. PLOS authors have the option to publish the peer review history of their article (what does this mean?). If published, this will include your full peer review and any attached files.

Reviewer #1: No

Reviewer #2: No

Reviewer #3: No

---

## [Author Response · Author response to Decision Letter 0]

10 Jan 2023

Reviewer #1:

Thank you for the opportunity to review this paper. Overall, this is a good paper. However, I have suggestions that can enhance the quality of this work. The authors may consider these suggestions to improve the paper. I wish the authors all the best with the revision.

This paper first introduces the constructivist view of knowledge and learning, discusses the shaping of knowledge and learning by contextual factors, and further explores the influence of context on absorptive capacity. It also performs a meta-analysis.

The introduction of the paper is well-written and it convincingly motivates the case for this study and explains why the study is needed.

The authors have covered much of the literature. However, some recent relevant studies have been overlooked. I provide below some recent studies that are relevant to the study. I suggest the authors look at the recent works of Naqushbandi and Jaisumudin the last 5 years to identify research that strengthens the arguments presented in this paper. Their works on absorptive capacity, knowledge management and organizational context can help the authors develop the theoretical foundation of the paper. The authors may include appropriately in their literature review and discussion the following recent studies:

#The linkage between open innovation, absorptive capacity and managerial ties: A cross-country perspective. Journal of Innovation & Knowledge, (2022). 7, 100167.

#Knowledge infrastructure capability, absorptive capacity and inbound open innovation: evidence from SMEs in France. Production Planning & Control, (2019). 30, 893-906.

#The interplay of leadership, absorptive capacity, and organizational learning culture in open innovation: Testing a moderated mediation model. Technological Forecasting and Social Change, (2018). 133, 156-167.

#Intervening role of realized absorptive capacity in organizational culture–open innovation relationship: Evidence from an emerging market. Journal of General Management, (2017). 42, 5-20.

#Managerial ties and open innovation: examining the role of absorptive capacity. Management Decision, (2016). 54, 2256-2276.

Typo on page 20 L 2 "3. theoretical hypothesis". Also, I suggest the authors remove the word "measurement throughout. Eg 3.1.1 Moderating effects of absorptive capacity measurements should be "3.1.1 Moderating effects of absorptive capacity" Same for Section 3.1.2

Some of the tables (eg table 6) cannot be fully read due to improper formatting.

The authors have done a good job at explaining the study's findings. However, relating the findings to the extant literature - which is the hallmark of a good discussion - is missing. Please update the discussion section to reflect how your findings are in line or in contrast with previous studies.

The authors should highlight the implications of the paper for theory and practice. This should be done in detail to highlight the contribution of the paper.

Some language-related errors and typos exist – please have the paper proofread/edited.

I see potential in this paper. The authors need to revise it based on the feedback above. I wish the authors all the best and look forward to reading an improved version of the study. All the best.

Comment1：However, some recent relevant studies have been overlooked. I provide below some recent studies that are relevant to the study. I suggest the authors look at the recent works of Naqushbandi and Jaisumudin the last 5 years to identify research that strengthens the arguments presented in this paper. Their works on absorptive capacity, knowledge management and organizational context can help the authors develop the theoretical foundation of the paper. The authors may include appropriately in their literature review and discussion the following recent studies

Response1：First of all, we are very grateful to the reviewers for providing references for this paper. We have carefully read the studies done by Naqushbandi and Jaisumudin and subsequently compared the commonalities between their studies and the present study. We found that Naqushbandi and Jaisumudin's study focused heavily on the specific contexts in which absorptive capacity comes into play. This fits well with the theme of this paper. Therefore, we used their study as a reference for this paper.

Comment2：Typo on page 20 L 2 "3. theoretical hypothesis". Also, I suggest the authors remove the word "measurement throughout. Eg 3.1.1 Moderating effects of absorptive capacity measurements should be "3.1.1 Moderating effects of absorptive capacity" Same for Section 3.1.2

Some of the tables (eg table 6) cannot be fully read due to improper formatting.

Response2：We corrected the error above and changed the subheading to “Moderating effects of absorptive capacity”.

Comment3: However, relating the findings to the extant literature - which is the hallmark of a good discussion - is missing. Please update the discussion section to reflect how your findings are in line or in contrast with previous studies.

The authors should highlight the implications of the paper for theory and practice. This should be done in detail to highlight the contribution of the paper.

Response3: We have revised the discussion section by adding a description of the implications for research and practice. In this section, we explore in detail the similarities and differences between the findings of this paper and those of existing studies, and present the contributions of this paper.

Comment4: Some language-related errors and typos exist – please have the paper proofread/edited.

Response4: We polished the language by the editing service to improve readability of the manuscript.

Reviewer #2:

This paper presents a qualitative theoretical discussion and a quantitative META analysis to examine the reasons for the apparent discrepancy between the empirical results of absorptive capacity and firm performance in existing studies. In the process of theoretical discussion, this paper finds that the core concept of absorptive capacity is based on cognitivism, and the process of absorptive capacity is based on mechanical cognitive information processing, which focuses on the internal cognitive structure and process of the subject. However, due to the dynamic and complex nature of social phenomena, the process model cannot effectively respond to the influence of contextual factors on their relationships. Therefore, this paper first introduces the constructivist view of knowledge and learning, discusses the shaping of knowledge and learning by contextual factors, and further explores the influence of context on absorptive capacity and constructs a model of the role of context on absorptive capacity and firm performance. Secondly, the paper analyzes the results of the existing empirical studies by META analysis and examines different contexts, and finds that the role of absorptive capacity on firm performance has obvious contextual characteristics, among which: country context, industry context, dynamic context, and firm size have significant moderating effects on absorptive capacity and firm performance. In addition, the research context also has an important impact on the relationship, including the way absorptive capacity and firm performance are measured and the type of data. In summary, this paper expands the meaning of absorptive capacity and examines the role of context in moderating absorptive capacity and firm performance. The abstract is too long. It should focus on the aim, the methods and the results. The methodology should further explained. Finally, the text should be English proofread because some sentences are not clear.

Comment1：The abstract is too long. It should focus on the aim, the methods and the results.

Response1: We reorganized the abstract section from scratch. Every effort was made to reduce the word count while keeping the abstract content focused on the objectives, methods, and results of the paper.

Comment2: The methodology should further explained.

Response2: This paper has been enriched with specific content in the research methods section to ensure that the Meta-analysis methods in this paper are understandable.

Comment3: Finally, the text should be English proofread because some sentences are not clear.

Response3: We polished the language by the editing service to improve readability of the manuscript.

Reviewer #3:

Absorptive capacity is an established construct that is frequently used as a moderator of the effect of external resource involvement on firm performance. The model chosen for the study does not examine this and examines the direct correlation with performance. This does not correspond to the definitional core of the theory. At the same time, this creates ambiguity of differentiation of AC from related management constructs such as innovativeness. Why was the approach chosen?

The relevance of the constructivism perspective to the study is not comprehensible. I recommend deleting this paragraph.

The focused moderators are sound. However, they are of limited conceptual value. It would be interesting to see other contextual variables, such as the entrepreneurial objective (e.g., exploration, exploitation), the type of external collaborators involved (e.g., firms, universities), and the form of collaboration....

The differences in the measurement of Absorptive Capacity are not comprehensible. Please, specify the two types of measurement investigated. What exactly was measured in the underlying studies? Currently, the exact definition of AC and thus the message of the meta-analysis is unclear. Alternatively, I suggest distinguishing unidimensional measurements /e.g. R&D intensity), multidimensional measurements, and subjective measurements in the analysis.

The measurement of performance only considers differences between subjective and objective measurements. More relevant would have been contextual distinctions between financial performance and technological performance, and between long-term and short-term performance.

The inclusion criteria of the studies are unclear. If I apply the search string (with the correct brackets) more than 6000 studies are included in the results only in web of science. Why has your search only identified 316 studies? Were only studies that measured financial performance included? Why no studies examining R&D performance? Some studies such as Lane et al. (2001), Flatten et al. (2011), Tzokas et al. (2015), or Melnychuk et. al. (2021) - just to name a few - are not included. Why not? The authors should provide a complete list of studies taken into account and not taken into account.

Some meta-analyses exist that examine absorptive capacity in the context of innovation, strategic alliances, and supply chains. How does the authors' meta-analysis distinguish itself from these?

Overall, the field of absorptive capacity is very established in the management literature. There are also some systematic reviews and meta-analyses available. In the current version of the study, it is not clear how the authors distinguish themselves from it. This is also evident in the very short discussion, which hardly considers conceptual implications from the analysis. What can practice and science learn from the analysis? PLOSONE is an interdisziplinary journal. As such this implication should also help other disciplines.

Comment1: Absorptive capacity is an established construct that is frequently used as a moderator of the effect of external resource involvement on firm performance. The model chosen for the study does not examine this and examines the direct correlation with performance. This does not correspond to the definitional core of the theory. At the same time, this creates ambiguity of differentiation of AC from related management constructs such as innovativeness. Why was the approach chosen?

Response1: Although absorptive capacity is often used as a mediating variable in other studies, performance has been one of the most important outcome variables in terms of the development of absorptive capacity theory itself. On the other hand, there are a large number of differing empirical results in studies on absorptive capacity and firm performance. Effectively clarifying the reasons for the emergence of differential results can help to explore the mechanism of the role of absorptive capacity on firm performance.

Comment2:The relevance of the constructivism perspective to the study is not comprehensible. I recommend deleting this paragraph.

Response2: In fact, constructivism is an important basis for this paper to propose context as a moderating variable between absorptive capacity and firm performance. Previous studies have treated the process model of absorptive capacity as a linear model, while treating knowledge as an objective fact. However, with the introduction of the constructivist perspective, the linear nature of absorptive capacity will change, while the context becomes an important variable of absorptive capacity acting on performance due to the change in knowledge characteristics.

Comment3: The focused moderators are sound. However, they are of limited conceptual value. It would be interesting to see other contextual variables, such as the entrepreneurial objective (e.g., exploration, exploitation), the type of external collaborators involved (e.g., firms, universities), and the form of collaboration....

Response3: Indeed, as the reviewer stated, factors such as entrepreneurial purpose, collaborators, and form of collaboration are very valuable research themes, but coding these variables can be very difficult for META analysis. For this study, it was difficult to code the purpose of entrepreneurship or collaborators for all the study sample in one study. This is because of the limited number of studies in this paper. Therefore, a conventional empirical study may be more appropriate for the exploration of the above variables.

Comment4: The differences in the measurement of Absorptive Capacity are not comprehensible. Please, specify the two types of measurement investigated. What exactly was measured in the underlying studies? Currently, the exact definition of AC and thus the message of the meta-analysis is unclear. Alternatively, I suggest distinguishing unidimensional measurements /e.g. R&D intensity), multidimensional measurements, and subjective measurements in the analysis.

The measurement of performance only considers differences between subjective and objective measurements. More relevant would have been contextual distinctions between financial performance and technological performance, and between long-term and short-term performance.

Response4: Basically, as you might expect, we divide the measurement of absorptive capacity into R&D-related one-dimensional measurements and process-related multidimensional measurements. The reason for this is that the measurement with R&D-related metrics treats absorptive capacity as a stock, while the process-related multidimensional scale treats absorptive capacity as a flow that can be changed quickly.

For performance measurement, same as your idea, we have tried to divide performance into long-term performance and short-term performance. But the current research is rather fragmented in its discussion of performance, and it is difficult for us to find a criterion to divide the performance measurement into long-term and short-term. So we still used the objective and subjective division.

Comment5: The inclusion criteria of the studies are unclear. If I apply the search string (with the correct brackets) more than 6000 studies are included in the results only in web of science. Why has your search only identified 316 studies? Were only studies that measured financial performance included? Why no studies examining R&D performance? Some studies such as Lane et al. (2001), Flatten et al. (2011), Tzokas et al. (2015), or Melnychuk et. al. (2021) - just to name a few - are not included. Why not? The authors should provide a complete list of studies taken into account and not taken into account.

Response5: Yes, the performance studied in this paper focuses on the financial performance or business performance of the firm. For R&D performance or innovation performance is not the scope of this paper. This is because the difference between absorptive capacity and firm performance is mainly reflected in financial performance. The current empirical studies have more consistent findings for both innovation performance and R&D performance.

For the sample selection section, we have revised the method section. We hope to make it clearer and easier to understand. Also, we will upload the list of included studies in the hope that it will solve the reviewers' questions.

Comment6: Some meta-analyses exist that examine absorptive capacity in the context of innovation, strategic alliances, and supply chains. How does the authors' meta-analysis distinguish itself from these?

Response6: First, the main questions addressed in this paper focus on absorptive capacity and firm performance. And this aspect still lacks quantitative tests conducted by META analysis. Secondly, this paper provides the basis for the hypothesis formulation of META analysis by conducting a lot of theoretical discussions before the meta-analysis.

---

## [Decision Letter · Decision Letter 1]

14 Feb 2023

Is absorptive capacity the "panacea" for organizational development? A META Analysis of Absorptive Capacity and Firm Performance from the Perspective of Constructivism

PONE-D-22-28180R1

Dear Dr. Pu,

We’re pleased to inform you that your manuscript has been judged scientifically suitable for publication and will be formally accepted for publication once it meets all outstanding technical requirements.

Kind regards,

Muhammad Ali, PhD

Academic Editor

PLOS ONE

Additional Editor Comments (optional):

Dear Authors

I am pleased to inform you that both reviewers have suggested acceptance of your paper for publication. Our editorial team will be in touch with you for the next steps. I wish you all the best!

Reviewers' comments:

Reviewer's Responses to Questions

**Comments to the Author**

1. If the authors have adequately addressed your comments raised in a previous round of review and you feel that this manuscript is now acceptable for publication, you may indicate that here to bypass the “Comments to the Author” section, enter your conflict of interest statement in the “Confidential to Editor” section, and submit your "Accept" recommendation.

Reviewer #1: All comments have been addressed

Reviewer #2: All comments have been addressed

2. Is the manuscript technically sound, and do the data support the conclusions?

Reviewer #1: Yes

Reviewer #2: Yes

3. Has the statistical analysis been performed appropriately and rigorously? 

Reviewer #1: Yes

Reviewer #2: Yes

4. Have the authors made all data underlying the findings in their manuscript fully available?

Reviewer #1: No

Reviewer #2: Yes

5. Is the manuscript presented in an intelligible fashion and written in standard English?

Reviewer #1: Yes

Reviewer #2: Yes

6. Review Comments to the Author

Reviewer #1: (No Response)

Reviewer #2: (No Response)

7. PLOS authors have the option to publish the peer review history of their article (what does this mean?). If published, this will include your full peer review and any attached files.

Reviewer #1: No

Reviewer #2: No

---

## [Editor Report · Acceptance letter]

16 Feb 2023

PONE-D-22-28180R1 

*Is absorptive capacity the "panacea" for organizational development?
A META Analysis of Absorptive Capacity and Firm Performance from the Perspective of Constructivism*

Dear Dr. Pu:

I'm pleased to inform you that your manuscript has been deemed suitable for publication in PLOS ONE. Congratulations! Your manuscript is now with our production department. 

Kind regards, 

on behalf of

Dr. Muhammad Ali 

Academic Editor

PLOS ONE